# Cost-utility analysis of LEGO based therapy for school children and young people with autism spectrum disorder: results from a randomised controlled trial

Han-I Wang ,[1] Barry Debenham Wright,[1,2] Matthew Bursnall,[3] Cindy Cooper,[3] Ellen Kingsley,[2] Ann Le Couteur,[4] Dawn Teare,[5] Katie Biggs ,[3] Kirsty McKendrick,[3] Gina Gomez de la Cuesta,[6] Tim Chater,[7] Amy Barr,[3] Kiera Solaiman,[3] Anna Packham,[3] David Marshall ,[8] Danielle Varley,[1] Roshanak Nekooi,[2] Simon Gilbody,[1] Steve Parrott[1]

**Correspondence to**
Dr Han-I Wang;
han-i.wang@york.ac.uk

## ABSTRACT

**Objectives** To assess the cost-effectiveness of LEGO-based therapy compared with usual support.

**Design** Cost-utility analysis alongside randomised control trial.

**Setting** Mainstream primary and secondary schools in the UK.

**Participants** 248 children and young people (CYP) with autism spectrum disorder (ASD) aged 7–15 years.

**Intervention** LEGO-based therapy is a group social skills intervention designed specifically for CYP with ASD. Through play, CYP learn to use the skills such as joint attention, sharing, communication and group problem-solving. CYP randomised to the intervention arm received 12 weekly sessions of LEGO-based therapy and usual support, while CYP allocated to control arm received usual support only.

**Main outcome measures** Average costs based on National Health Service (NHS) and personal social services perspective and quality-adjusted life years (QALYs) measured by EQ-5D-Y over time horizon of 1 year were collected during the trial. Incremental cost-effectiveness ratio (ICER) was calculated, and non-parametric bootstrapping was conducted. The uncertainty around the ICER estimates was presented using cost-effectiveness acceptability curve (CEAC). A set of sensitivity analyses were conducted to assess the robustness of the primary findings.

**Results** After adjustment and bootstrapping, on average, CYP in LEGO-based therapy group incurred less costs (incremental cost was −£251 (95% CI −£752 to £268)) and gained marginal improvement in QALYs (QALYs gained 0.009 (95% CI −0.008 to 0.028)). The CEAC shows that the probability of LEGO-based therapy being cost-effective was 94% at the willingness-to-pay threshold of £20 000 per QALY gained. Results of sensitivity analyses were consistent with the primary outcomes.

**Conclusion** Compared with usual support, LEGO-based therapy produced marginal reduction in costs and improvement in QALYs. Results from both primary and sensitivity analyses suggested that LEGO-based therapy was likely to be cost-effective.

## Strengths and limitations of this study

► The first economic evaluation study of LEGO-based therapy.
► The data are from a large sample size trial in autism spectrum disorder, and the method followed published best-practice guidance.
► The study accounts for the costs measured from a range of perspectives and the quality-adjusted life years measured by different instruments.
► Resource use data were collected retrospectively and may be affected by inaccurate recall.
► Further model-based evaluation is required to assess long-term cost-effectiveness of LEGO-based therapy.

**Trial registration number** ISRCTN64852382.

## BACKGROUND

Autism spectrum disorders (ASDs) are a group of lifelong developmental conditions defined by impairments in social interactions, communication skills, and presence of restricted and stereotypical behaviours.[1] It is estimated that around 120 000 children and young people (CYP) (1.1% of total CYP) in the UK have a diagnosis of ASD.[2–4] CYP with ASD have differing health and quality of life outcomes to neurotypically developing people, and over their lifetime this is likely to have a financial impact on their families or carers. In the UK, the annual cost of supporting children with ASD has been estimated at £3.1-3.4 billion (in 2011 value) with higher values when there is associated intellectual disability. The main cost driver of the annual support cost was special education (47%) followed by parental productivity loss

(12%). Medical services accounted for only 4% of the total.[5]

Due to the substantial financial burden borne by the healthcare system, education system and families of CYP with ASD, the Lancet Psychiatry Commission emphasises the need to not only focus on the effectiveness of mental health services but also on their economic benefits.[6] However, with the evolving intervention landscape of ASD, only two economic evaluation studies were found for CYP with ASD.[7] One investigated the cost-effectiveness of a communication intervention[8] and another investigated an early intervention programme.[9] These preliminary studies suggested that early intervention for children with suspected ASD was cost-effective, but communication-focused therapy for preschool children with ASD was not, with more research being recommended.

LEGO-based therapy[10] is a group-based social skills intervention specifically designed for CYP with ASD, which does not rely on adult-led teaching of skills. It has become very popular in the UK with many local authorities now recommending its use in schools.[11] Despite the growing interest and substantial economic burden of ASD, no economic evaluation study has been done for LEGO-based therapy. Hence, the aim of this study was to access the cost-effectiveness of LEGO-based therapy. This paper reports the economic evaluation results of LEGO-based therapy for CYP with ASD alongside the Investigating SOcial Competence and Isolation in children with Autism taking part in LEGO-based therapy clubs In School Environments (I-SOCIALISE) trial.[12]

## METHODS
### Trial design and participants
This economic evaluation was embedded in the I-SOCIALISE trial, a multisite, pragmatic, two-arm, school-level cluster randomised controlled trial for CYP with ASD. Details of I-SOCIALISE trial have been published elsewhere.[12] In short, CYP between the ages of seven and 15 years with a diagnosis of ASD were recruited from mainstream primary and secondary schools in the North of England between October 2018 and May 2019. Parents/guardians and schools were invited to speak on the phone or face-to-face to discuss the eligibility of CYP in their school and their potential involvement in the study. CYP were included in the study if they met study inclusion criteria, which included aged between seven and 15 years with a clinical diagnosis of ASD, a score of 15 or higher on the Social Communication Questionnaire (SCQ), the ability to understand simple instructions, no serious impairments which would prevent participation and were attending mainstream schools in the north of England (see Appendix 1). CYP in schools that were allocated by remote randomisation to the intervention arm received a 1-hour session of LEGO-based therapy in school once per week for the 12-week period. On average, around three CYP were in each session. The decision about number of sessions and the duration per session were based on recommendations of the coauthor and experienced LEGO therapy trainer (Gina Comez de la Cuesta), the published training manual and in line with previous studies on school-based intervention (such as Social Stories[13]). There was some flexibility in weekly delivery to accommodate school timetables. CYP also received usual support, while CYP in schools allocated to the control arm received usual support only. Usual support includes any support the CYP with ASD was receiving at the time from school, general practitioners (GPs) or other professionals. The usual support from school includes the support from specialist teaching teams for autism as well as other interventions such as the picture exchange communication system, visual supports and timetables and social stories. To investigate the efficacy and cost-effectiveness of LEGO-based therapy while controlling for any impact obtained from usual support, CYP received LEGO-based therapy, and usual support was compared with usual support alone, rather than to another similar intervention. Informed consent and baseline measurements were obtained and completed prior to school randomisation. All CYP were followed up 20 and 52 weeks after randomisation and completed further outcome measures. A flowchart of the study is found in Appendix 2.

### Intervention
LEGO-based therapy is a group social skills intervention designed specifically for CYP with social communication difficulties such as ASD. CYP build LEGO models together in small groups with light facilitation by a trained adult.[14 15] The CYP work together taking on one of three roles: the engineer, who reads the LEGO set instructions; the supplier, who finds the correct pieces according to the instructions from the engineer and the builder, who builds the model with the pieces from the supplier and the instructions from the engineer. Key elements of the intervention include the use of CYP-led collaborative building and learning through play, which promote the learning and use of such skills as joint attention, sharing, communication and group problem-solving. The trained adult takes a guiding role rather than an explicitly directive one, allowing the CYP to work together and solve their own challenges. Group rules and rewards are used to foster motivation and engagement in social interactions.

### Outcomes
The health outcomes for the current study were quality adjusted life years (QALYs) measured by the EQ-5D-Y (3 L proxy version)[16] and the Child Health Utility 9D (CHU-9D).[17] EQ-5D-Y (3 L proxy version) is a 5-item with three-severity-level questionnaire that allows a proxy person (ie, parent/guardian) to complete the measure for CYP. The EQ-5D-Y instrument comprises five dimensions (mobility, looking after themselves, doing usual activities, having any pain or discomfort and feeling worried or sad) and has been shown to be a reliable and valid instrument for use in CYP and adolescents.[16] The CHU-9D is

a CYP-completed 9-item questionnaire comprising nine dimensions: worried, sad, pain, tired, annoyed, school-work/homework, sleep, daily routine and able to join in activities.[17]

Individual-level responses to EQ-5D-Y and CHU-9D were used to estimate utilities based on UK population valuation sets.[18 19] A utility represents a CYP's health state and is constrained between 0 and 1, where 0 refers to death and 1 perfect health. The estimated utilities at baseline and follow-up were further joint using the area under the curve approach[20] to calculate QALYs.

## cost measurement
Two cost perspectives were considered in this study: (1) a National Health Service (NHS) and personal social service (PSS) perspective, which included costs related to healthcare (including hospital-based services, such as inpatient stays, outpatient visits and emergency care and services outside a hospital setting, such as GP visits, Child and Adolescent Mental Health Services (CAMHS) and services provided by allied health professionals (eg, community paediatrician)) and social services (including social care worker, home care worker, family support worker and Helpline (eg, Samaritans)) and (2) a societal perspective, which additionally included costs of education-related services, parental out-of-pocket expenses (such as childcare and private courses) and parental productivity costs (time off work due to child's autism) without taking social charges (any payments or contributions for social benefits) into consideration.

## Cost of the intervention
Cost of the intervention included the cost of training and the cost of delivering LEGO-based therapy. Training costs were measured by the time spent by the trainer and included travel costs and the cost of materials used in the training. Costs associated with delivering the LEGO-based therapy were measured based on the time spent by facilitators to plan and conduct sessions and included relevant overhead costs. All relevant data were collected using the tailored questionnaires completed by the study team and therapists.

## Cost of the service use
Service use was collected using the tailored questionnaires (completed by the parent/guardian and separately by an associated teacher of each CYP in the study who knew the CYP well), which was originally developed based on Barrett's study[21] and has been successfully adapted for use in school-based trials.[22 23] The parent/guardian-completed questionnaire captured data on the use of health and social services, school-based services (including school-based health services, general and intervention support), parental private expenses and parental productivity costs. Teacher-completed questionnaires captured any school-based interventions/support and the implications of a CYP's behaviour on school resource.

Service use was multiplied by unit costs to arrive at total cost in each arm. Unit costs of health and social service use were obtained from published sources (ie, Reference Cost[24], Personal Social Services Research Unit 2018[25] and Prescription Cost Analysis[26]), national survey (ie, Childcare Costs Survey 2018[27]) and government departments(ie, Department for Education 2018[28] and Green Book 2018[29]). Privately paid services were separately estimated via market prices, while productivity losses were calculated using the human capital approach, which involves multiplying time off work by UK average salary.[30]

All the costs were expressed in 2018 UK sterling. Discounting of costs and QALYs was not applied, as the study time horizon was 1 year.

## Missing data
All eligible CYP who had both utility and cost data at any time point were referred to as complete cases. While, the eligible CYP who had missing utility and cost data but had complete baseline assessments were referred to as base case. The identified missing utility and cost data were imputed using multiple imputation method via chained equations.[31] Imputation was based on trial arm, age, gender, study site and utility scores and SCQ scores at baseline.

## Statistical and economic analyses
The primary analysis was to calculate incremental cost-effectiveness ratio (ICER) based on the costs from NHS/PSS perspective and the QALYs measured by EQ-5D-Y. To account for uncertainty around ICER and imbalanced utility and costs at baseline, seemingly unrelated regression equations (SURE) that adjusted SEs for clustering and controlled for baseline utility,[32] costs, age, gender and SCQ scores were bootstrapped 5000 times. The SURE approach was recommended by Glick and colleagues,[20] which considers the distribution of the dependent variable and any correlation found between cost and QALY outcomes. While non-parametric bootstrap resampling method was suggested by Briggs and colleagues,[33] as the distribution of regression residuals was likely to be skewed.[34] The 5000 bootstrapped iterations were represented graphically on the cost-effectiveness plane (CE-plane), and the cost-effectiveness acceptability curve was generated by plotting the probability of the intervention being cost-effective over a range of willingness-to-pay (WTP) thresholds.[35] The calculated ICERs were against the national WTP threshold of £20 000–£30 000 per QALY gained to decide whether the LEGO-based therapy is cost-effectiveness.[36]

To assess the robustness of our findings, a set of sensitivity analyses were conducted. First, a cost-utility analysis (CUA)[35] using complete cases was conducted to assess the impact of the missing data. Second, a CUA was performed from a NHS/PSS and education perspective to account for the economic impact from the education system. Third, a CUA was performed from a societal perspective to account for all the economic impact outside the

NHS/PSS perspective, including parental productivity costs. Finally, a CUA that used the CHU-9D instead of the EQ-5D-Y to estimate QALYs based on the UK population tariff[19] was conducted to assess the impact of outcome measurement instrument.

All analyses were performed on an intention-to-treat basis using Stata V.16 (StataCorp, College Station, Texas).

### Ethical approval and informed consent
This study was funded by the National Institute for Health Research Public Health Research (PHR) programme (PHR15/49/32).[2] The written informed consent was obtained from parents on behalf of their child. Children assented to be part of the groups and did not proceed if they were not willing.

### Patient and public involvement
No patient involved.

## RESULTS
### Participants
A total of 284 CYP with ASD were recruited in the trial. After removing 34 ineligible CYP and 2 CYP who were not eligible for multiple imputation due to missing baseline utilities, 248 CYP with ASD were available for primary analysis (126 were allocated to LEGO-based therapy and 122 to usual support). This sample constitutes the base-case

group. On the other hand, only 139 (56.0%) CYP had both EQ-5D and resource use (from the NHS/PSS perspective) data at all three data collection time points. This sample constitutes the complete-case group. Overall, 27.8% of cost or QALY measurements was missing and were imputed for primary analysis.

Descriptive statistics of CYP's baseline characteristics for both complete-case and base-case are presented in table 1. As shown, more than third-quarters of the CYP in the LEGO-based therapy and the usual care arms were men, and more than 50% of the CYP in both arms were in primary school age (ranging from 7 to 11 years old). Differences in the SCQ scores at the baseline were marginal across arms and samples. Overall, the baseline characteristics are consistent across samples (base-case and complete-case).

### Costs
On average, the estimated intervention cost per session per CYP was £6.5 (£2.45 for training and £4.05 for intervention delivery). The main cost driver of training costs was trainer fees, while the main cost drivers of intervention delivery costs were the costs for preparation and delivery the intervention and the costs for LEGO materials (Appendix 2).

In terms of service costs, the average total service costs over 52 weeks to the NHS providers (after imputation)

**Table 1** Baseline characteristics by trial arm

| | Base case (n=248) | | Complete case (n=139) | |
|---|---|---|---|---|
| Baseline characteristics | LEGO-based therapy (N=126) | Usual support (N=122) | LEGO-based therapy (N=80) | Usual support (N=59) |
| Gender, n (%) | | | | |
| Male | 101 (80.2%) | 91 (74.6%) | 68 (85.0%) | 43 (72.9%) |
| Age (years), n (%) | | | | |
| 7–11 | 83 (65.9%) | 79 (64.8%) | 54 (67.5%) | 43 (72.9%) |
| 11–15 | 43 (34.1%) | 43 (35.2%) | 26 (32.5%) | 16 (27.1%) |
| Mean (SD) | 9.7 (2.3) | 9.8 (2.2) | 9.6 (2.2) | 9.6 (2.2) |
| Year from diagnosis | | | | |
| Mean (SD) | 3.4 (2.7) | 3.6 (2.8) | 3.2 (2.4) | 3.6 (3.0) |
| SCQ scores | | | | |
| Mean (SD) | 25.1 (5.2) | 24.2 (5.2) | 24.9 (5.1) | 24.1 (5.0) |
| EQ-5D | | | | |
| Mean (SD) | 0.79 (0.11) | 0.76 (0.11) | 0.79 (0.12) | 0.77 (0.11) |
| Site, n (%) | | | | |
| Leeds | 37 (29.4%) | 38 (31.2%) | 31 (38.8%) | 18 (30.5%) |
| Sheffield | 70 (55.6%) | 67 (54.9%) | 34 (42.5%) | 31 (52.5%) |
| York | 19 (15.1%) | 17 (13.9%) | 15 (18.7%) | 10 (17.0%) |
| Number of intervention sessions | | | | |
| Mean (SD) | 10.3 (2.3) | – | 10.5 (2.2) | – |

SCQ, Social Communication Questionnaire.

**Table 2** Average costs of service use in 1 year by trial arm

| | Base case | | Complete case | |
|---|---|---|---|---|
| | LEGO-based therapy (n=126), £ (95% CI) | Usual support (n=122), £ (95% CI) | LEGO-based therapy (n=80), £ (95% CI) | Usual support (n=59), £ (95% CI) |
| **NHS and PSS** | **524 (372 to 675)** | **678 (427 to 928)** | **618 (428 to 808)** | **752 (420 to 1,083)** |
| Community-based services | | | | |
| CAMHS-related | 77 (40 to 114) | 233 (37 to 428) | 117 (50 to 184) | 267 (19 to 516) |
| Non-CAMHS-related | 115 (79 to 151) | 99 (69 to 130) | 120 (78 to 161) | 107 (67 to 148) |
| Hospital-based services | | | | |
| Mental health-related | 20 (6 to 33) | 45 (11 to 79) | 19 (4 to 33) | 53 (-1 to 107) |
| Non-mental health-related | 60 (29 to 92) | 86 (37 to 136) | 79 (30 to 128) | 89 (31 to 147) |
| Medications | | | | |
| Mental health-related | 195 (115 to 275) | 129 (74 to 185) | 211 (121 to 301) | 142 (75 to 208) |
| Non-mental health-related | 57 (18 to 97) | 85 (25 to 145) | 73 (22 to 124) | 93 (19 to 167) |
| **Education system-related** | **1204 (949 to 1,458)** | **1437 (1,082 to 1,792)** | **1388 (989 to 1,787)** | **1633 (1,041 to 2,224)** |
| School-based health | 164 (62 to 267) | 88 (20 to 156) | 182 (48 to 316) | 100 (13 to 186) |
| Intervention support | 712 (496 to 927) | 948 (645 to 1250) | 793 (458 to 1128) | 1070 (615 to 1526) |
| General support | 327 (242 to 413) | 401 (262 to 541) | 368 (245 to 492) | 473 (237 to 709) |
| **Private expenses** | **211 (129 to 293)** | **317 (189 to 445)** | **192 (105 to 280)** | **329 (171 to 487)** |
| Childcare | 211 (129 to 293) | 317 (189 to 445) | 192 (105 to 280) | 329 (171 to 487) |
| **Productivity** | **95 (57 to 132)** | **114 (64 to 164)** | **104 (62 to 146)** | **111 (53 to 170)** |
| Parental productivity loss | 95 (57 to 132) | 114 (64 to 164) | 104 (62 to 146) | 111 (53 to 170) |
| **Total costs** | **2033 (1710 to 2357)** | **2546 (2087 to 3005)** | **2278 (1775 to 2781)** | **2819 (2123 to 3515)** |

Community-based services: health services provided outside of a hospital setting, including services provided by GPs, by applied health professionals (community nurse, community paediatrician, occupational therapist, physiotherapist and Speech and Language therapist for non-CAMHS-related services, and child psychiatrist, child psychotherapist, child psychologist, clinical psychologist, mental health nurse and primary mental health worker for CAMHS-related services) and by social services (social care worker, home care worker, family support worker and Helpline (eg, Samaritans))
Hospital-based services: health services provided in a hospital setting, including inpatient stays, outpatient visits and emergency services.
Childcare included paid childcare, after school club, religious club and special clubs for autism children
CAMHS including child psychiatrist, child psychotherapist, child psychologist, clinical psychologist, mental health nurse and primary mental health worker.
CAMHS, Child and Adolescent Mental Health Services; GP, general practitioner; NHS, National Health Service; PSS, personal social service.

were £524 (95% CI £428 to 808) for the LEGO-based therapy arm compared with £678 (95% CI £427 to 928) for the usual care arm. The cost difference is larger when the societal perspective was considered, as CYP in the LEGO-based therapy arm incurred less costs across all the perspectives. The largest cost differences occurred in education-related services followed by healthcare and social services. It is worth noting that some of cost differences were likely to have been driven by high-cost cases. For instance, in complete-case, higher average cost of CAMHS-related services in the usual care arm was driven by two high-cost cases, and higher average cost of school-based health-related services in the LEGO-based therapy arm was driven by one high-cost case. These high-cost cases were kept in the analysis, as they were plausible. However, due to the high-cost cases, the cost differences need to be interpreted with caution. A more detailed overview on

the service costs over 52 weeks and the resource use are presented in table 2 and Appendix 3, respectively.

## Outcomes

Table 3 shows the mean EQ-5D-Y (3 L proxy) and CHU-9D utility scores between the two arms of the trial at each time point when scores were not imputed (complete case) and when scores were imputed (base case). As shown, in both arms, there was no significant change in EQ-5D-Y or CHU-9D utility scores from baseline to 52 weeks. The fluctuations between baseline and 20 weeks and between 20 weeks and 52 weeks were small in both the base and the complete cases. After calculation using the area under the curve approach, it is found that the LEGO-based therapy produced marginally higher mean QALYs (0.03 QALYs) compared with the usual support regardless of the instrument used. Further details for the

**Table 3** Average EQ-5D-Y and CHU-9D utility scores by trial arm

| Time point | Base case | | Complete case | |
| --- | --- | --- | --- | --- |
| | LEGO-based therapy (n=126), mean (95% CI) | Usual support (n=122) mean (95% CI) | LEGO-based therapy (n=80), mean (95% CI) | Usual support (n=59) mean (95% CI) |
| EQ-5D-Y | | | | |
| Baseline | 0.79 (0.77 to 0.81) | 0.76 (0.74 to 0.79) | 0.79 (0.77 to 0.81) | 0.77 (0.74 to 0.79) |
| 20 weeks | 0.78 (0.76 to 0.81) | 0.76 (0.74 to 0.78) | 0.79 (0.76 to 0.81) | 0.75 (0.74 to 0.79) |
| 52 weeks | 0.79 (0.76 to 0.81) | 0.76 (0.74 to 0.79) | 0.80 (0.77 to 0.82) | 0.75 (0.73 to 0.80) |
| Total QALYs | 0.79 (0.77 to 0.80) | 0.76 (0.74 to 0.78) | 0.79 (0.77 to 0.81) | 0.76 (0.74 to 0.79) |
| CHU-9D | | | | |
| Baseline | 0.83 (0.80 to 0.85) | 0.81 (0.79 to 0.84) | 0.84 (0.81 to 0.87) | 0.83 (0.80 to 0.86) |
| 20 weeks | 0.84 (0.82 to 0.86) | 0.80 (0.78 to 0.83) | 0.83 (0.80 to 0.86) | 0.78 (0.74 to 0.82) |
| 52 weeks | 0.83 (0.81 to 0.85) | 0.80 (0.77 to 0.83) | 0.81 (0.78 to 0.84) | 0.81 (0.77 to 0.85) |
| Total QALYs | 0.83 (0.82 to 0.85) | 0.80 (0.78 to 0.82) | 0.82 (0.80 to 0.85) | 0.80 (0.76 to 0.83) |

CHU-9D, Child Health Utility 9D; QALYs, quality-adjusted life years.

responses of EQ-5D and CHU-9D in each domain are found in Appendix 4 and Appendix 5, respectively.

### CUA (primary analysis)

After accounting for uncertainty and unbalanced baseline utility and characteristics, on average, CYP with ASD receiving LEGO-based therapy incurred £251 (95% CI –£268 to £752) less costs from the NHS/PSS perspective and gained 0.009 (95% CI –0.008 to 0.028) extra QALYs measured by EQ-5D-Y than those having usual support. The bootstrapped ICER results are presented in figure 1A, and the probabilities of LEGO-based therapy being cost-effective over a range of WTP threshold are presented in figure 1B. As shown, the simulated estimates were largely below the threshold line, and the probability of LEGO-based therapy being cost-effective is 94% at the WTP threshold of £20 000. The findings suggest that the LEGO-based therapy was likely to be cost-effective, although the incremental costs and QALYs were marginal.

### Sensitivity analysis

Results of sensitivity analyses are detailed in Appendix 6. The mean incremental cost and QALY estimates from the complete case were along the line of the based-case scenario, yielding a negative cost per QALY gained. Sensitivity analyses using a societal perspective to measure costs and the CHU-9D to measure QALYs were also conducted. In both sensitivity analyses, the ICER pairs lay below the recommended National Institute for Health and Care Excellence (NICE) threshold (£20 000–30,000/QALY gained), and the majority of the bootstrapped estimates sat in the bottom right quadrant (figure 2), suggesting that LEGO-based therapy was likely to be cost-effective.

### DISCUSSION
### Principal findings

To the best of our knowledge, this is the first study evaluating the cost-effectiveness of LEGO-based therapy for

CYP with ASD. Compared with usual care, the LEGO-based therapy marginally decreased the service use costs and increased the QALYs from the NHS/PSS perspective. This is evident in both primary and sensitivity analyses,

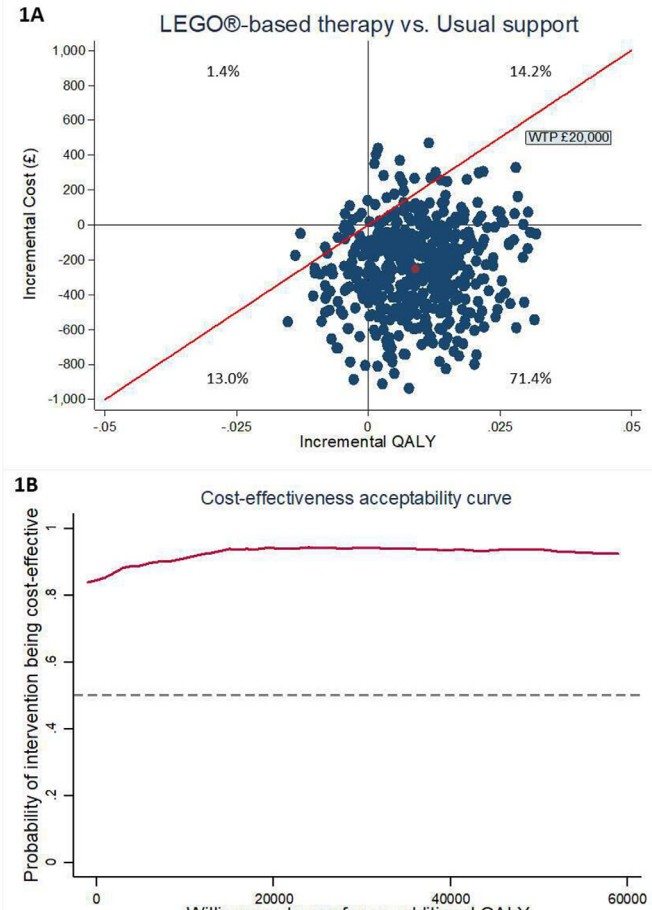

**Figure 1** Cost-effectiveness plane and CEAC of primary analysis. CEAC, cost-effectiveness acceptability curve; QALYs, quality-adjusted life years; WTP, willingness-to-pay.

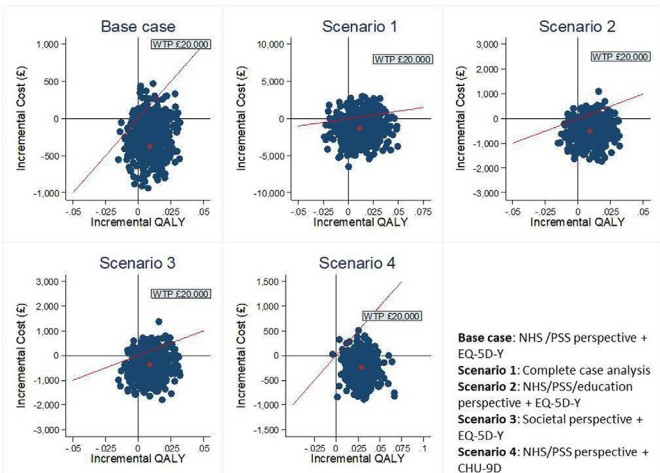

**Figure 2** Cost-effectiveness planes of sensitivity analyses. NHS, National Health Service; PSS, personal social service; QALYs, quality-adjusted life years; WTP, willingness-to-pay.

which considered costs derived from various perspectives and QALYs measured by different instruments.

### Implications of study

The average QALYs measured by both the EQ-5D-Y (proxy version) and the CHU9D for those in the intervention arm were higher compared with the control arm. Although the differences are small, it is observed that the bootstrap estimates plotted on the CE planes for both primary and sensitivity analyses were mainly in the bottom right quadrant. This indicates that after taking uncertainty into consideration and adjusting for the imbalanced baseline utility, the differences in QALYs remain positive. It is also worth noting that such differences were unaffected regardless who filled the questionnaire. Both the parent-completed (EQ-5D) and CYP-completed (CHU9D) questionnaires showed the same positive differences.

The study also shows a reduction in average total NHS/PSS costs (although with wide confidence intervals), particularly through attendance at CAMHS. As mentioned in section Results: Costs, the lowered CAMHS costs found in the LEGO-based therapy arm was more related to a small number of CYP receiving high levels of high-tariff CAMHS support in the control arm (usual support) than the intervention arm rather than a general drop across the whole group. Such a finding indicates that CYP who had co-occurring emotional and behavioural problems seemed to be receiving approximately similar support from CAMHS across both trial arms, and LEGO-based therapy did not overshadow the needs of CAMHS support. It was also found that, at the baseline, CYP in LEGO-based therapy arm had higher frequency of CAMHS support compared with control arm. Given the high threshold for receiving CAMHS support, it is likely that the CYP in LEGO-based therapy arm had more severe needs at baseline. However, after the intervention, CYP in LEGO-based therapy arm had similar support from CAMHS as those in control arm (see above) suggesting some amelioration effect or a reduced need for such high-level support. Based on the

literature of general research on CAMHS referrals, such reduction could be because the LEGO-based therapy improves comorbid problems of CYP and consequently leads to a reduced likelihood of referral to CAMHS[37] or stops the CYP being seen by CAMHS.[38] However, it is also possible that the school-based LEGO-based therapy improves a CYP's social skills and leads to less distress or conflict and may subsequently reduce the likelihood for referral to CAMHS[39] because of reduced levels of teacher and parent/guardian concern. Research to date, however, suggests higher parental anxiety tends to reduce referral rates not increase them in many circumstances.[40]

The reduction in school intervention costs was also observed in this study. One possible explanation is that CYP in a LEGO-based therapy intervention might be less likely to be put forward for other interventions (eg, the social use of language programme[41]). Another possible reason might include a belief by a parent/guardian that an active intervention is happening, and so for this reason taking part in another intervention at the same time is not necessary. While both possible explanations might be valid, there was evidence that schools reported a wide range of other interventions being received by CYP with ASD, including Social Stories, visual schedules, 1:1 mentoring and others. Whether LEGO-based therapy reduced certain type of interventions but increases other type of interventions remains unclear. Further investigations would be needed to explore this.

Finally, the bootstrapping results on the CE-plane (figures 1A and 2) demonstrate the dispersion of iterations. It is observed that the 95% CIs for incremental costs and incremental QALYs were wide, both in primary and sensitivity analyses. This indicated high levels of uncertainty around the estimate of the incremental costs and QALYs and, consequently, wide CIs of the estimated ICERs. The phenomenon may be due to the small average cost reduction and small mean QALY gained, but large variation among the CYP. This could be also because the EQ-5D-Y instrument can be less responsive or sensitive to small changes in mental health.[42] Although the CIs for the ICERs were wide, the LEGO-based therapy remains highly likely to be cost-effective, as the majority of cost-QALY pairs were below the £20 000 threshold (figures 1 and 2).

### Strengths and limitations

This is the first study evaluating the cost-effectiveness of LEGO-based therapy and one of only a few economic evaluation studies for CYP with ASD.[8 9] Such a study is important because there is a growing popularity of LEGO_based therapy in the UK. Furthermore, since detailed resource use in school was able to be collected via teacher-completed questionnaire, our study managed to capture the cost difference in school in a more granular manner and reflect the reality in school better. Additionally, our study accounts for the costs measured from a range of perspectives and the QALYs measured by different instruments. The approach not only ensures the

robustness of our findings but also can help policymakers from different sectors to make informed decision. This is particularly true in the UK setting, as organisations such as the Department for Education, the Department of Health and Social Care and the local authorities are working together to ensure CYP with special educational needs and disabilities (SEND) properly supported based on the SEND Code of Practice 2014[43] and the Children and Families Act 2014 [44]. Some other considerations beyond cost effectiveness, such as acceptability and equality of access, may be also taken into account by decision makers. Findings of strong acceptability to schools, children, CYP and their families were reported elsewhere,[45] whereas equality of access would need further exploration in the future, as at present, we do not have sufficient data to undertake any form of statistical analysis. Finally, this study has a large sample size compared with other similar trials in ASD[8 46] and is one of the few ASD intervention studies to date follow-up to 1 year.[8 47] This would make our results more generalisable and robust compared with the similar study with small sample size or shorter term follow-up.

Despite the strengths mentioned above, our study was subjected to a few limitations. First, funding sources for a few types of staff were not always clear. For example, speech and language therapists can be funded by NHS, schools or local authorities. Such diversity causes difficulties when it comes to costing and reporting the results, as detailed information about funders for each member of staff involved was unavailable. Several assumptions have been made based on service locations and published guidelines (ie, the unit costs of Health and Social Care from PSSRU 2018) for costing. Hence, the summarised cost results for different perspectives need to be treated with caution. However, both arms were treated the same way and the overall costs (from the societal perspective) should remain robust. Second, a small number of high but plausible values were observed in cost estimations. Although the values did not affect the cost-effectiveness conclusions (data not shown), such values can potentially bias the cost reduction results of LEGO-based therapy. Hence, care should be taken when interpreting the cost estimates. It is none-the-less important to include this real-world data and be aware of this for future studies. Third, the calculated intervention costs might have been underestimated. This is because several items associated with training and intervention sessions were not costed, due to data constraints. These included opportunity costs of trainee time, opportunity costs of school venue for delivering interventions, recruitment cost if intervention rolled out and supervision costs. However, this is unlikely to have affected the results of the dominance of the intervention over usual support, as these costs are considered to be small. This is especially the case after the allocating to every CYP for every session. Further research on the exploration and measurement of the costs with considerations is desirable. Finally, our study used SURE to model the uncertainty around the incremental costs and QALYs and account for their correlation. Alternatively, costs and QALYs can be modelled separately using generalised linear models without considering the correlation. It is beyond the scope of our study to compare the two methods. However, further research on the method comparison and their impacts on the results are desirable in order to draw robust conclusions.

### Future work
Our study measured the short-term cost-effectiveness of LEGO-based therapy on CYP with ASD over 1-year follow-up. For the long-term cost-effectiveness of LEGO-based therapy, a modelled-based economic evaluation study would be desirable to allow life time cost-effectiveness and children's lost productivity during adulthood to be measured. Further research is also needed on exploring potential impacts on other outcomes such as academic achievement or quality of life of other family members. In future research, it would be also helpful to explore whether a longer duration of intervention (eg, a full school year) or more frequent sessions (eg, two times a week) would further improve outcomes while remaining cost-effective.

### CONCLUSION
This study demonstrates the potential cost-effectiveness of delivering LEGO-based therapy to CYP with ASD in mainstream school settings. The findings will be of interest to NHS health and social care providers, local authorities, families and community professionals including school staff members.

**Author affiliations**
[1]Department of Health Sciences, University of York, York, UK
[2]COMIC Research Team, Leeds and York Partnership NHS Foundation Trust, Leeds, UK
[3]Clinical Trials Research Unit, ScHARR, The University of Sheffield, Sheffield, UK
[4]Sir James Spence Institute, Newcastle University, Newcastle upon Tyne, UK
[5]Population Health Sciences Institute, Newcastle University, Newcastle upon Tyne, UK
[6]Play Included a Community Interest Company, Cambridge, UK
[7]Liverpool Clinical Trials Centre, University of Liverpool, Liverpool, UK
[8]Centre for Reviews and Dissemination, University of York, York, North Yorkshire, UK

**Acknowledgements** We gratefully thank all families participating in the study. We acknowledge invaluable support and guidance from our trial steering committee (Professor David Cottrell (University of Leeds), Karen Watson (PPI), Dr Sue Fletcher-Watson (University of Edinburgh), Dr Fiona Warren (University of Exeter), Dr Michael Morton (University of Glasgow), Alison Thompson (Leeds and York Partnership NHS Foundation Trust)), and our data monitoring and ethics committee (Dr Zoe Hoare (Bangor University), Dr David Simms (Bradford District Care NHS Foundation Trust), John O'Dwyer (University of Leeds)). We also gratefully acknowledge the hard work, support and advice from the following: PPI representatives The Young Dynamos research advisory group, the National Autistic Society, Karen Watson (PPI TSC member), the York Youth Council and several individuals including Tina Hardman and Ann LcLaren. Participant screening and data collection Sarah Jacob-Eshtan, Lisa Hackney, Rebecca Hargate, Sam Bennett, Emma Sellers, Holly Taylor, Alix Smith, Katie Sutherland, Richard Campbell, Rachel Hodkinson, Megan Garside, Jane Blackwell, Pavithra Kumar, Jennifer Lomas Administrative and clerical support Sharon Bird, Rita Lynch, Daniel Gottschalk, Katy Harmston, Thasamia Akhtar, Louise Turner, Heather Dakin Sponsor Study monitoring Tahir Idrees LEGO® based therapy expertise Elinor Brett, Abigail Dodson Support on data management

concerns Emily Turton Support on ethical and governance issues. Sinead Audsley, Stephen Holland The authors would also like to thank Daniel LeGoff for information about LEGO® based therapy in the USA and both the Leeds and York Partnership NHS Foundation Trust and the University of York for supporting the Child Oriented Mental Health Intervention Centre (COMIC) and the University of Sheffield CTRU in taking this research from an idea to a full trial. The research team are aware that the LEGO® name is a registered trademark and will follow the fair use policy in regard to the LEGO® brand throughout the duration of the trial. The team have been in discussion with LEGO, and they have agreed for the use of this term for the project and its outputs, but not over the longer term. After PPI work, we have paired the term LEGO® based therapy with the new term Play Brick Therapy, which we suggest is used henceforth.

**Contributors** H-IW and SP were the trial health economists and analysed the data with H-IW taking the lead in writing the manuscript. BDW, CC, ALC, DT, KB, GGdIC, DM, DV and SG conceived the study idea and designed the project. MB and DT were the study statisticians and involved in the analysis and interpretation of the data. EK and KM managed the trial, and TC, AB, KS, AP and RN were involved in the acquisition and management of the data. All authors contributed to data interpretation and have read and approved the final manuscript, and H-IW is responsible for the overall content as the guarantor.

**Funding** This work was supported by the NIHR under its Programme Grants for Public Health Research (PHR) (Grant Reference Number PHR15/49/32).

**Disclaimer** The views and opinions expressed by authors in this publication are those of the authors and do not necessarily reflect those of the NHS, the NIHR, NETSCC, the PHR programme or the Department of Health and Social Care.

**Competing interests** The research team was aware that the LEGO name is a registered trademark and followed the fair use policy in regard to the LEGO brand throughout the duration of the trial. GGdIC co-authored the LEGO-based therapy manual, which formed the basis of the intervention delivered in the trial. The co-authors of the manual have given us full permission to use the manual without license and to develop an abridged version. They have also stated their support for us in writing our own version and will become co-authors on any future publications. GGdIC has also agreed for the team to adapt the fidelity checklist used in her previous study. GGdIC is a Director of Play included a community interest company that offers training and resources for interventions involving play bricks for children. We have provisional agreement with Jessica Kingsley Publishers who have expressed interest in publishing the abridged manual. However, we are not tied to them as a publisher. There are no other financial and/or competing interests to declare.

**Patient consent for publication** Not applicable.

**Ethics approval** This study involves human participants and ethics approval has been issued by the University of York Research Ethics Committee (18/HRA/0101). Participants gave informed consent to participate in the study before taking part.

**Provenance and peer review** Not commissioned; externally peer reviewed.

**Data availability statement** Data are available upon reasonable request. No additional data available. However, Stata code that used for this study is available from the corresponding author upon reasonable request.

**ORCID iDs**
Han-I Wang http://orcid.org/0000-0002-3521-993X
Katie Biggs http://orcid.org/0000-0003-4468-7417
David Marshall http://orcid.org/0000-0001-5969-9539

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
