## [Reviewer comments · BMJ Open]

ARTICLE DETAILS

TITLE (PROVISIONAL)	Cost-utility analysis of LEGO® based therapy for school children and young people with autism spectrum disorder: results from a randomised controlled trial
AUTHORS	Wang, Han-I; Wright, Barry; Bursnall, Matthew; Cooper, Cindy; Kingsley, Ellen; Le Couteur, Ann; Teare, Dawn; Biggs, Katie; McKendrick, Kirsty; Gomez de la Cuesta, Gina; Chater, Tim; Barr, Amy; Solaiman, Kiera; Packham, Anna; Marshall, David; Varley, Danielle; Nekooi, Roshanak; Gilbody, Simon; Parrott, Steve

VERSION 1 – REVIEW

REVIEWER	Sampaio, Filipa Uppsala Universitet, Department of Public Health and Caring Sciences
REVIEW RETURNED	04-Sep-2021

GENERAL COMMENTS	Comments to the author This study investigates the cost-effectiveness of a LEGO based therapy plus usual support compared to usual support only for children and young people with ASD in the UK. The authors mention that LEGO based theory has a growing popularity in the UK and that results from this evaluation would be valuable to decision makers in diverse sectors. Nonetheless, economic evaluations of interventions targeting ASD are scarce and the present study adds great value to the field of ASD research. I have a few comments and questions to the authors. Methods 2.1. Trial design and participants Page 5, lines 12-17 – The authors described that children and young people in schools randomized to the intervention arm received 12 sessions weekly. How long is each session? And how many children received the intervention on average per group? They also mention that children received usual support, GPs and other professionals. What is included in usual support at school? 2.3. Outcomes Page 5, lines 48-49 - Please clarify in text whether the EQ5D Y used is the 3L or the 5L version (it is only mentioned later on in the results and should be mentioned in the methods section). 2.4 cost measurement Cost of service use – page 6 – please describe in text what types of services are included in health and social services collected via the questionnaire.
--

	Page 7, line 7 – were social fees included in the estimation of productivity losses? 2.5. Missing data Page 7 - What was the proportion of missing data imputed? 2.6. Statistical and economic analyses Page 7 – Could the authors clarify why the EQ5D-Y was preferred for the base case analysis to the CHU9D? Additionally, the authors have used SUR regression models to analyze costs and qalys based on its ability to account for correlation between costs and qalys. I wonder, however, why the authors chose this method instead of for instance generalized linear models, which accounts for different distributions other than normal for cost and qaly outcomes. Were cost and qaly outcomes in the present study normally distributed? Also, did the SUR models accommodate clustering? Please clarify. Results Page 10 – table 2 – it is unclear from the table what is included in community based services and hospital based services. Table notes should be connected to the items in the table. 3.4. Cost-utility analysis Figure 1A and 1B - would be useful for the reader to be able to see the proportion of iterations in each quadrant in the CE plane figures. Also the dispersion of the iterations should be discussed, related to the wide confidence intervals of the difference in costs and qalys between the trial arms and its impact on cost effectiveness conclusions 3.5. Sensitivity analysis Page 12, line 17 - Suggest replacing fourth quadrant by more standardly used terminology, ie, lower right or south east quadrant Discussion 4.3. Strengths and limitations Page 13 – lines 57-58 – The authors mention that the different perspectives used in the study not only ensure the robustness of our findings but also can help policy makers from different sectors to make informed decision. I am curious about which policy makers would benefit most from these study results in the UK context. Are schools the ones deciding whether to fund LEGO based therapy? Would other considerations beyond cost effectiveness, such as equity of access to such therapy, sustainability of funding etc need to be taken into account for these decision makers? Adding a few discussion points around these aspects would strengthen the discussion. 4.4 Future work Page 14 - Potential impacts on other outcomes such as academic achievement or quality of life of other family members would also be interesting to explore.
--	---

REVIEWER	Hu, Xiaoyi Beijing Normal University, Department of Special Education
REVIEW RETURNED	20-Oct-2021

GENERAL COMMENTS	1. Please define what is usual support. Why we compare this with lego? 2. please provide a rationale for 12 sessions of lego therapy 3. there is much individual difference among students with ASD in terms of verbal, social and behavioral characteristics. Please introduce how the sample is suitable for this intervention.
--

VERSION 1 – AUTHOR RESPONSE

Comments from Reviewer #1

This study investigates the cost-effectiveness of a LEGO based therapy plus usual support compared to usual support only for children and young people with ASD in the UK. The authors mention that LEGO based theory has a growing popularity in the UK and that results from this evaluation would be valuable to decision makers in diverse sectors. Nonetheless, economic evaluations of interventions targeting ASD are scarce and the present study adds great value to the field of ASD research. I have a few comments and questions to the authors.

The authors would like to thank the reviewer for the thoughtful comments and efforts towards improving our manuscript. A point-by-point response to Reviewer 1's specific comments is provided below.

Comment 1:

2.1. Trial design and participants

Page 5, lines 12-17 – The authors described that children and young people in schools randomized to the intervention arm received 12 sessions weekly. How long is each session? And how many children received the intervention on average per group? They also mention that children received usual support, GPs and other professionals. What is included in usual support at school?

Response 1:

The authors thank the reviewer for the comments and have revised the text as suggested.

(a) The details about the intervention have been added in section 2.1. Trial design and participants as follows:

“... CYP in schools that were allocated by remote randomisation to the intervention arm received a one-hour session of LEGO® based therapy in school once per week for the 12-week period. ...”

(b) The details about the number of children received the intervention on average per session have been added in section 2.1. Trial design and participants as follows:

“...On average, around three CYP were in each session. ...”

(c) The details about the usual care from school have been further defined as suggested in section 2.1. Trial design and participants as follows:

“... The usual support from school includes the support from specialist teaching teams for autism as well as other interventions such as the Picture Exchange Communication System, visual supports and timetables and Social Stories. ...”

Comment 2:

2.3. Outcomes

Page 5, lines 48-49 - Please clarify in text whether the EQ5D Y used is the 3L or the 5L version (it is only mentioned later on in the results and should be mentioned in the methods section).

The authors thank the reviewer for the suggestion. The details about the EQ-5D-Y have been added as suggested in 2.3 Outcomes as follows:

“The health outcomes for the current study were quality adjusted life years (QALYs) measured by the EQ-5D-Y (3L proxy version) (16) and the Child Health Utility 9D (CHU-9D) (17). EQ-5D-Y (3L proxy version) is a five-item with three-severity level questionnaire that allows a proxy person (i.e. parent/guardian) to complete the measure for CYP. ...”

Comment 3:

2.4 cost measurement

Cost of service use – page 6 – please describe in text what types of services are included in health and social services collected via the questionnaire.

The authors thank the reviewer for the suggestion and have added details about types of services included in section 2.4 cost measurement as follows:

“... 1) a NHS and personal social service (NHS/PSS) perspective, which included costs related to healthcare (including hospital-based services, such as inpatient stays, outpatient visits and emergency care, and services outside a hospital setting, such as GP visits, Child and Adolescent Mental Health Services, and services provided by allied health professionals (e.g. community paediatrician)) and social services (including social care worker, home care worker, family support worker, and Helpline (e.g. Samaritans)), ...”

Comment 4:

Page 7, line 7 – were social fees included in the estimation of productivity losses?

The social fees were not included in the estimation of productivity losses, and this has been clarified in the text as follows:

“... 2) a societal perspective, which additionally included costs of education-related services, parental out-of pocket expenses (such as childcare and private courses), and parental productivity costs (time off work due to child’s autism) without taking social charges (any payments or contributions for social benefits) into consideration.”

Comment 5:

2.5. Missing data

Page 7 - What was the proportion of missing data imputed?

The authors agree with the reviewer that the proportion of missing data was not very clearly described in section 3.1 Participants and have revised the text as follows:

“..., 248 CYP with ASD were available for primary analysis (126 were allocated to LEGO® based therapy and 122 to usual support). This sample constitutes the base-case group. On the other hand, only 139 (56.0%) CYP had both EQ-5D and resource use (from the NHS and PSS perspective) data at all three data collection time points. This sample constitutes the complete-case group. Overall, 27.8% of cost or QALY measurements were missing and were imputed for primary analysis.”

Comment 6:

2.6. Statistical and economic analyses

Page 7 – Could the authors clarify why the EQ5D-Y was preferred for the base case analysis to the CHU9D? Additionally, the authors have used SUR regression models to analyze costs and qalys based on its ability to account for correlation between costs and qalys. I wonder, however, why the authors chose this method instead of for instance generalized linear models, which accounts for different distributions other than normal for cost and qaly outcomes. Were cost and qaly outcomes in the present study normally distributed? Also, did the SUR models accommodate clustering? Please clarify.

(a) The authors thank the reviewer for the comment regarding the model selection. The seemingly unrelated regression equation (SURE) is one of the commonly used methods for trial-based economic evaluations (Willan 2006). Following the study protocol, the current study used SURE to account for correlation between costs and QALYs. This is also the approach taken by many studies (Heller 2017, Ramanan 2019). However, the authors agree with the reviewer that generalised linear models (GLMs) could have been used for modelling the costs, although it would mean that no correlation is considered. For information, the QALYs are approximately normally distributed in this study. It is outside the scope of this study to compare the results of the two methods. However, the authors have highlighted this issue in the limitations as follows:

“... Finally, our study used SURE to model the uncertainty around the incremental costs and QALYs and account for their correlation. Alternatively, costs and QALYs can be modelled separately using generalised linear models without considering the correlation. It is beyond the scope of our study to compare the two methods. However, further research on the method comparison and their impacts on the results are desirable in order to draw robust conclusions.”

Willan AR. Statistical analysis of cost-effectiveness data from randomized clinical trials. *Expert Rev Pharmacoecon Outcomes Res.* 2006;6(3):337–46.

Heller S, White D, Lee E, et al. A cluster randomised trial, cost-effectiveness analysis and psychosocial evaluation of insulin pump therapy compared with multiple injections during flexible intensive insulin therapy for type 1 diabetes: the REPOSE Trial. *Health Technol Assess.* 2017 Apr;21(20):1-278. doi: 10.3310/hta21200.

Ramanan AV, Dick AD, Jones AP, et al. Adalimumab in combination with methotrexate for refractory uveitis associated with juvenile idiopathic arthritis: a RCT. *Health Technol Assess.* 2019 Apr;23(15):1-140. doi: 10.3310/hta23150.

(b) The authors agree with the reviewer that the description about the SURE can be further clarified. The authors have revised the text in section 2.6 Statistical and economic analyses as follows:

“... To account for uncertainty around ICER and imbalanced utility and costs at baseline, seemingly unrelated regression equations (SURE) that adjusted standard errors for clustering and controlled for baseline utility (32), costs, age, gender, and SCQ scores were bootstrapped 5,000 times. ...”

Comment 7:

Page 10 – table 2 – it is unclear from the table what is included in community based services and hospital based services. Table notes should be connected to the items in the table.

The authors thank the reviewer for spotting the error. The authors have revised the footnote of Table 2 as follows:

“Community-based services: health services provided outside of a hospital setting, including services provided by GPs, by applied health professionals (community nurse, community paediatrician, occupational therapist, physiotherapist, and Speech and Language therapist for non-CAMHS related services, and child psychiatrist, child psychotherapist, child psychologist, clinical psychologist, mental health nurse, and Primary mental health worker for CAMHS related services) and by social services (social care worker, home care worker, family support worker, and Helpline (e.g. Samaritans)).

Hospital-based services: health services provided in a hospital setting, including inpatient stays, outpatient visits and emergency services.”

Comment 8:

3.4. Cost-utility analysis

Figure 1A and 1B - would be useful for the reader to be able to see the proportion of iterations in each quadrant in the CE plane figures.

The authors thank the reviewer for the suggestion and have added the percentage figure for each quadrant in Figure 1A.

Comment 9:

Also the dispersion of the iterations should be discussed, related to the wide confidence intervals of the difference in costs and qalys between the trial arms and its impact on cost effectiveness conclusions

The authors agree with the reviewer's comment and have added the discussion about the dispersion of the iterations in section 4.2 Implications of study as follows:

“Finally, the bootstrapping results on the cost-effectiveness plane (Figure 1A and 2) demonstrate the dispersion of iterations. It is observed that the confidence intervals for incremental costs and incremental QALYs were wide, both in primary and sensitivity analyses. This indicated high levels of uncertainty around the estimate of the incremental costs and QALYs and, consequently, wide confidence intervals of the estimated ICERs. The phenomenon may be due to the small average cost reduction and small mean QALY gained, but large variation among the CYP. This could be also because the EQ-5D-Y instrument can be less responsive or sensitive to small changes in mental health (42). Although the confidence intervals for the ICERs were wide, the LEGO® based therapy remains highly likely to be cost-effective, as the majority of cost-QALY pairs were below the £20,000 threshold (Figure 1 and 2).”

Comment 10:

3.5. Sensitivity analysis

Page 12, line 17 - Suggest replacing fourth quadrant by more standardly used terminology, ie, lower right or south east quadrant

The authors thank the reviewer for the suggestion and have revised the text as follows:

“... , and the majority of the bootstrapped estimates sat in the bottom right quadrant (Figure 2), ...”

Comment 11:

4.3. Strengths and limitations

Page 13 – lines 57-58 – The authors mention that the different perspectives used in the study not only ensure the robustness of our findings but also can help policy makers from different sectors to make informed decision. I am curious about which policy makers would benefit most from these study results in the UK context. Are schools the ones deciding whether to fund LEGO based therapy? Would other considerations beyond cost effectiveness, such as equity of access to such therapy, sustainability of funding etc need to be taken into account for these decision makers? Adding a few discussion points around these aspects would strengthen the discussion.

The authors thank the reviewer for the suggestion and have added some discussion points in section 4.3. Strengths and limitations as follows:

“...This is particularly true in the UK setting, as organisations such as the Department for Education (DfE), the Department of Health and Social Care (DH), and the local authorities are working together to ensure CYP with special educational needs and disabilities (SEND) properly supported based on the SEND Code of Practice 2014 (43) and the Children and Families Act 2014 (44). Some other considerations beyond cost effectiveness, such as acceptability and equality of access may be also taken into account by decision makers. Findings of strong acceptability to schools, children, CYP and their families were reported elsewhere (45), whereas equality of access would need further exploration in the future, as at present we do not have sufficient data to undertake any form of statistical analysis. ...”

Comment 12:

4.4 Future work

Page 14 - Potential impacts on other outcomes such as academic achievement or quality of life of other family members would also be interesting to explore.

The authors thank the reviewer for the suggestion and have revised the text as follows:

“...Further research is also needed on exploring potential impacts on other outcomes such as academic achievement or quality of life of other family members.”

Comments from Reviewer #2

The authors thank the reviewer for the positive and constructive feedback.

Comment 1:

Please define what is usual support. Why we compare this with lego?

The authors thank the reviewer for the suggestions.

(a) The definition of usual support has been added in section 2.1 Trial design and participants as follows:

“... Usual support includes any support the CYP with ASD was receiving at the time from school, general practitioners (GPs) or other professionals. The usual support from school includes the support from specialist teaching teams for autism as well as other interventions such as the Picture Exchange Communication System, visual supports and timetables and Social Stories. The usual support from school includes the support from specialist teaching teams for autism as well as other interventions such as the Picture Exchange Communication System, visual supports and timetables and Social Stories. ...”

(b) The rationale of comparing usual support with LEGO® based therapy has been added in section 2.1 Trial design and participants as follows:

“...To investigate the efficacy and cost-effectiveness of LEGO® based therapy whilst controlling for any impact obtained from usual support, CYP received LEGO® based therapy and usual support were compared to usual support alone, rather than to another similar intervention. ...”

Comment 2:

Please provide a rationale for 12 sessions of lego therapy.

The authors thank the reviewer for the suggestion and have added the rationale for 12 sessions of LEGO® based therapy in section 2.1 Trial design and participants as follows:

“... The decision about the number of sessions and the duration per session were based on recommendations of the co-author and experienced LEGO –therapy trainer (Gina Comez de la Cuesta) , the published training manual and in line with previous studies on school based interventions (such as Social Stories (13). ...”

(13): Wright B, Marshall D, Adamson J, Ainsworth H, Ali S, Allgar V, et al. Social Stories™ to alleviate challenging behaviour and social difficulties exhibited by children with autism spectrum disorder in mainstream schools: design of a manualised training toolkit and feasibility study for a cluster randomised controlled trial with nested qualitative and cost-effectiveness components. *Health Technol Assess (Rockv)*. 2016;20(6):1–258.

The authors also added the comment about number of sessions in section 4.4 Future work as follows:

“... In future research, it would be also helpful to explore whether a longer duration of intervention (e.g. a full school year) or more frequent sessions (e.g. twice a week) would further improve outcomes while remaining cost-effective”

Comment 3:

There is much individual difference among students with ASD in terms of verbal, social and behavioral characteristics. Please introduce how the sample is suitable for this intervention.

The authors thank the reviewer for the comment and agree that students with ASD are likely to have a range of individual differences and are also likely to have additional social behaviour and mental health co-occurring problems. The authors have revised the text in section 2.1 Trial design and participants as follows:

“... Parents/guardians and schools were invited to speak on the phone or face-to-face to discuss the eligibility of CYP in their school and their potential involvement in the study. CYP were included in the study if they met study inclusion criteria which included aged between seven and 15 years with a clinical diagnosis of ASD, a score of 15 or higher on the SCQ, the ability to understand simple instructions, no serious impairments which would prevent participation, and were attending mainstream schools in the north of England (see Appendix 1). ...”

VERSION 2 – REVIEW

REVIEWER	Sampaio, Filipa Uppsala Universitet, Department of Public Health and Caring Sciences
REVIEW RETURNED	06-Dec-2021
GENERAL COMMENTS	I would like to thank the authors for thoroughly answering my comments. They have done a wonderful job. I have no further comments.